# Preclinical Evaluation of an Innovative Bone Graft of Marine Origin for the Treatment of Critical-Sized Bone Defects in an Animal Model

Rafael Otero-Pérez [1,2,3,*], María Permuy [4,5], Estefanía López-Senra [1], Miriam López-Álvarez [1], Mónica López-Peña [4,5], Julia Serra [1,3], Antonio González-Cantalapiedra [4,5], Fernando M. Muñoz [4,5] and Pío González [1,3]

1 CINTECX, New Materials Group, Campus de Vigo, Universidade de Vigo, As Lagoas, Marcosende, 36310 Vigo, Spain; eslopez@uvigo.es (E.L.-S.); miriammsd@uvigo.es (M.L.-Á.); jserra@uvigo.es (J.S.); pglez@uvigo.es (P.G.)
2 Traumatology and Orthopedic Surgery Unit, Hospital POVISA, 36211 Vigo, Spain
3 Galicia Sur Health Research Institute (IIS Galicia Sur), SERGAS-UVIGO, 36213 Vigo, Spain
4 Department Veterinary Clinical Sciences, Faculty of Veterinary, University of Santiago de Compostela, 27002 Lugo, Spain; permuy@ibonelab.com (M.P.); monica.lopez@usc.es (M.L.-P.); antonio.cantalapiedra@usc.es (A.G.-C.); fernandom.munoz@usc.es (F.M.M.)
5 iBoneLab SL, Avenida da Coruña 500, 27003 Lugo, Spain
* Correspondence: rotero@povisa.es; Tel.: +34-986-413144 (Work) or +34-626-417078 (Home)

**Abstract:** Autogenous cancellous bone graft is the current gold standard of treatment for the management of bone defects since it possesses the properties of osteoinduction, osteoconduction, and osteogenesis. Xenografts and synthetic grafts have been widely reported as available and low-cost alternatives, which retain good osteoconductive and mechanical properties. Given the rich biodiversity of ocean organisms, marine sources are of particular interest in the search for alternative bone grafts with enhanced functionalities. The purpose of this paper is to assess the biocompatibility of a marine-derived bone graft obtained from shark tooth, which is an environmentally sustainable and abundant raw material from fishing. This research presents the findings of a preclinical trial—following UNE-EN ISO 10993—that induced a critical-sized bone defect in a rabbit model and compared the results with a commercial bovine-derived bone graft. Evaluation by micro-computed tomography and histomorphometric analysis 12 weeks after implantation revealed good osseointegration, with no signs of inflammatory foreign body reactions, fibrosis, or necrosis in any of the cases. The shark tooth-derived bone graft yielded significantly higher new bone mineral density values (54 ± 6%) than the control (27 ± 8%). Moreover, the percentage of intersection values were much higher (86 ± 8%) than the bovine-derived bone graft (30 ± 1%) used as control. The area of occupancy by bone tissue in the test material (38 ± 5%) also gave higher values than the control (30 ± 6%). The role of physicochemical properties, biphasic structure, and composition on the stimulation of bone regeneration is also discussed.

**Keywords:** marine bone graft; preclinical trial; calcium phosphate; osseointegration; biomaterial

## 1. Introduction

Bone grafts offer a mechanical support and a biological function in the treatment of several pathologies, among these high-grade open fractures with bone loss, high-energy trauma, non-union or delayed union, resection of bone tumors, infections requiring extensive bone debridement, and secondary bone defects [1]. Opinions diverge as to the most appropriate bone replacement treatment, when considered from both a clinical and an economic perspective.

There is a paucity of clinical evidence and controlled studies comparing different techniques to inform strategies and treatments for critical-sized bone defects. Current

therapeutic approaches include distraction osteogenesis, first reported by Ilizarov in the 1950s [2], as well as Masquelet's more recently developed induced membrane technique [3]. Furthermore, the use of bone grafts—autografts, allografts, xenografts, and synthetic grafts—has been widely communicated. Autogenous cancellous bone graft possesses the properties of osteoinduction, osteoconduction, and osteogenesis, making it the current gold standard of treatment for the management of bone defects. Additionally, it is non-immunogenic and does not carry the risk of transmissible infection [4,5]. However, the morbidity associated with harvesting the amount of graft required to fill the defect is substantial [6–8].

In orthopedic surgery, there are situations where synthetic bone substitutes have the same clinical results as autografts, but with the advantage that they do not consume surgical time and are not limited in availability. In particular, the development of bone grafts of natural origin obtained from non-human species has received much attention in recent decades. Xenografts are mainly derived from bovine, porcine, equine, and marine sources. These bone grafts can provide structural support and osteoconduction; however, osteogenesis and osteoinduction are not possible. The fabrication process is based on heating and chemical treatments leading to the production of hydroxyapatite (HA) and/or other calcium phosphate compounds. Then, the organic and osteogenic components of the bone are removed, although the potential risk of disease transmission (bovine spongiform encephalopathy) is not completely avoided [9]. Xenografts of bovine origin have given good results in dental and maxillofacial surgeries, but contradictory results have been achieved in orthopedic surgery, in particular foot and ankle surgery [10,11].

Xenografts obtained from marine sources are of particular interest given the rich biodiversity of ocean organisms with a similar structure to the human bone, which makes them suitable bone graft biomaterials with osteoconductive properties. Corals are among the most studied marine-derived biomaterials. Coral-based biomaterial is mainly calcium carbonate and can be chemically transformed into HA by a hydrothermal conversion to increase the strength of the coral skeletons [12]. Current clinical evidence is limited to well-contained voids in dental and maxillofacial surgery, and to fill bony voids or gaps that do not affect bone stability in limbs and pelvis [13,14].

In recent years, other marine species have been investigated for use in bone tissue engineering: nacre, seashells (foraminifera; bivalve mollusks such as oysters and mussels), sponge skeletons, diatom frustules, sea urchin spines, cuttlefish bone, and other fish bones such as tuna [15]. Calcium carbonate predominates in most of these sources, except for certain sponge skeletons namely Demonspongiae and Hexactinellida and diatom frustules, which are based on silicon compounds [16].

Marine sources with a calcium phosphate composition are also under investigation. One of these, the subject of this study, is shark tooth from two commercial species, *Isurus oxyrinchus* and *Prionace glauca*. Shark tooth is gaining increasing recognition as a promising, eco-friendly source of bioapatites: it is an abundant by-product of fishing, is environmentally sustainable, and disease transmission risk from the marine ecosystem has not been reported. Preliminary in vitro [17] and in vivo evaluations [18] of these marine-derived grafts have confirmed good results for bone regeneration.

In this paper, we present the results of a preclinical trial of this marine-origin bone graft obtained from shark teeth, by means of a critical-sized bone defect test in an animal model. We carried out the biological evaluation and assessed biocompatibility according to UNE-EN ISO 10993: 2007 part 6. The purpose of this study was to gather initial findings on the potential of shark tooth-derived bone graft for the treatment of critical-sized bone defects, and to identify areas of focus for future research.

## 2. Material and Methods

### 2.1. Shark Tooth-Derived Bone Graft Processing Method

The marine-origin bone graft under evaluation was obtained from shark teeth of the *Prionace glauca* and *Isurus oxyrinchus* species provided by COPEMAR S.A. (Vigo, Spain).

The manufacturing method was similar to that described by López-Álvarez et al. [17,18], involving pyrolysis up to 1100 °C for 12 h, followed by grinding and sieving to select the desired grain size fraction, in this case granules with diameters between 0.5 and 1.0 mm. Finally, the graft material was packed in laminar flow cabins and sterilized by a 25 kGy dose of gamma radiation.

To determine the composition and structure of the shark tooth-derived bone graft, physicochemical characterization was performed using inductively coupled plasma optical emission spectrometry (ICP-OES, using a Perkin Elmer Optima 4300 DV), complemented by ion chromatography (DIONEX ICS-3000), Fourier-transform Raman spectroscopy (FT-Raman, using a Bruker RFS100 equipped with an Nd:YAG laser), and X-ray diffraction (with a Siemens D-5000 diffractometer).

Bio-Oss® (Geistlich Pharma AG, Wolhusen, Switzerland), composed of lyophilized anorganic bovine bone granules with a granule size ranging from 0.25 to 1.0 mm, was used as a control bone graft. It is a reference xenograft used in dental bone regeneration worldwide and has granules of a similar shape and size to those in the test sample (i.e., the shark tooth-derived bone graft).

### 2.2. Animal Model

Ten male New Zealand White rabbits were supplied by a qualified laboratory animal supply center (La Granja cunícola San Bernardo, S.L., Tulebras, Navarra), and were quarantined for 3 weeks prior to commencement of the study. They were aged between 18 and 21 weeks and weighed between 3.7 and 4.0 kg.

The animals were housed in individual enriched cages in the Animal Experimentation Service Facility of the University of Santiago de Compostela (Lugo, Spain) following ethical approval (procedure code: 03/18/LU-002) as a randomized controlled trial with one inter-subject control. Conditions such as temperature (15–21 °C), humidity, air renewal, and cycle time (12 h alternating cycles of light and dark) were controlled according to Annex II of Directive 86/609/EEC. The animals were fed with dry feed prepared in the form of pellets, sterilized and specially manufactured for the feeding of laboratory animals. The center's head of animal welfare, along with staff and the researchers, watched over the health of the animals throughout the experiment. There were no unexpected deaths.

After a quarantine, animals were premedicated with ketamine 25 mg/kg/IM (Ketamidor® 100 mg/mL; Richter pharma AG, Wels, Austria), medetomidine 50 μg/kg/IM (Sededorm® 1 mg/mL; Vetpharma Animal Health SL, Barcelona, Spain), and buprenorphine 0.03 mg/kg (Bupaq® 0.3 mg/mL; Richter pharma AG). The anesthesia was maintained by the inhalation of an $O_2$ and 2% isoflurane (Vetflurane; Virbac SA, Caro, France) mixture using a facemask.

After aseptic preparation of the skin, a lateral approach of the distal femur was performed, affecting the skin and subcutaneous tissue to subsequently dissect the musculature and expose the surface of the lateral femoral condyle with a periosteal elevator. A 6 mm defect was created using a trephine connected to a surgical motor INTRAsurg® 300 (KaVo, Biberach, Germany). Once the biomaterial was placed, in the lateromedial direction, the wound was sutured using 4–0 vicryl in deep planes and 3–0 nylon for the skin. The control biomaterial was placed in the contralateral limb following the same procedure. The 2 types of biomaterial were randomly assigned a total of 5 times to both the left and right sides based on a computer-generated randomization list.

After wound closure, the skin was soaked with povidone solution. After the surgical intervention, atipamezole (0.15 mg/kg/IM; Nosedorm® 5 mg/mL; Vetpharma Animal Health SL) was administered to reverse the effects of the medetomidine. The animals were monitored and returned to their accommodation, where they remained throughout the experimental period.

After the surgeries, pain was controlled with buprenorphine (0.01–0.03 mg/kg/IM, Bupaq® 0.3 mg/mL; Richter pharma AG) for 3 days and with meloxicam (0.2 mg/kg/SC, Metacam, Boehringer) for 5 days. As a prophylactic antibiotic, we used enrofloxacin

(5 mg/kg/12 h, Syvaquinol® 10% Oral Solution; Syva labs) for 3 weeks. In addition, the condition of the wounds was evaluated twice a week, and the cages were cleaned each time.

A bone graft was placed in each distal femur of all animals. In other words, a total of 20 bone grafts were placed in the 10 animals—2 different bone grafts per animal. To avoid errors due to the implantation side, 5 bone grafts of each type (test or control) were placed in the left femurs, and 5 of each type in the right femurs. The trial period used was 12 weeks.

The animals were sacrificed with an overdose of intravenous sodium pentobarbital (Dolethal 200 mg/mL, Vetoquinol) after sedation with ketamine 25 mg/kg + medetomidine 50 μg/kg + buprenorphine 0.03 mg/kg. Subsequently, the distal femurs were recovered by dissection and the use of a bone saw. Specimens were immediately immersed in a 10% buffered formalin solution for a minimum of 2 weeks until the time of processing. For identification, we used a code consisting of a number and a letter. The number indicated the animal (from 1 to 10) and the letter the side of the femur (right-R and left-L). All bone grafts were recovered.

### 2.3. Histological Preparation

Once fixed, the samples were dehydrated in increasing ethanol concentrations (80%, 96%, 100%, and 100%) under agitation. The inclusion was carried out using a photopolymerizable resin based on glycolmethacrylate at increasing concentrations (Technovit 7200 VLC, Kulzer, Germany). Polymerization took place through the use of high-intensity blue light with specific equipment. The undecalcified blocks obtained were prepared for analysis by microtomography and were subsequently sectioned for histology.

### 2.4. Micro-Computed Tomography

The samples were scanned using a high-resolution Micro-CT Skyscan 1172 scanner (Bruker MicroCT NV, Kontich, Belgium) located at the A Graña Marine Biology Station in Ferrol, A Coruña (University of Santiago de Compostela). The parameters used for scanning were the following: the X-ray source was set at 100 kV and 100 μA with a voxel size of 13.58 μm, using a 0.5 mm Al/0.08 mm Cu filter. The scanning was performed over a 360° rotation acquiring images every 0.4°. Once scanned, the images were reconstructed with NRecon® software (Bruker MicroCT NV, Kontich, Belgium) using the algorithm described by Feldkamp (Feldkamp, Davis, and Kress, 1984). In all cases the parameters were optimized to reduce image noise and possible artifacts. They were then reconstructed with histogram parameters (dynamic range) from 0 to 0.038 for all images, regardless of the material used. Reconstructed images were evaluated with Data Viewer® software (Bruker MicroCT NV, Kontich, Belgium) once the implant was perfectly aligned. For the analysis of the percentage of biomaterial in the volume of interest (VOI), 2 different VOIs were defined:

1.  VOI 1 (large VOI or 6 mm VOI). As the defect made in the animals was 6 mm, this distance was taken as a reference, creating a circular VOI of 6 mm in diameter and placing the bone graft in the center of that circle.
2.  VOI 2 (small VOI). A cylindrical VOI was manually profiled around the whole contour of the material.

Additionally, the percentage of intersection was defined, corresponding to the percentage of bone graft that is in contact with bone. VOI 1 (6 mm diameter) was used for this analysis. The percentage of binarized pixels was determined as material that was in contact with binarized pixels as bone, using CTAn® software (Bruker MicroCT NV, Kontich, Belgium).

### 2.5. Histological and Histometric Analysis

After the microtomography analysis, the samples were subjected to sectioning and microgrinding techniques using machinery from the Exakt system (Exakt Apparatebau,

Hamburg, Germany). The polymerized blocks were prepared, obtaining 2 sections per sample, which were reduced to a thickness of about 40 µm. The thin sheets were stained with Levai-Laczkó dye, which makes it possible to distinguish between lamellar and newly formed bone, and also to maintain the matrix-cellular bone architecture that is essential for the subsequent histomorphometric evaluation.

The histological evaluation was performed by semi-quantitative evaluation techniques according to Annex E, table E.3 of ISO 10993-6: 2007. This evaluation estimates the degree of compatibility of the bone graft in relation to a recognized control. Two independent and experienced observers blinded to the treatments performed the scoring. From the quantification of cellular observations, by counting the number of various cells per high-power field, the irritant assessment was recorded as non-irritant (for values from 0.0 up to 2.9), slight irritant (from 3.0 up to 8.9), moderate irritant (9.0 up to 15.0), and severe irritant (>15).

The histometric analysis was performed by trained staff who did not know what material was being analyzed. The images were captured using an optical microscope (BX51, Olympus, Tokyo, Japan) connected to a digital color camera (DP71, Olympus, Tokyo, Japan) equipped with a motorized plate (Märzhäuser, Steindorf, Germany). The images were aligned and connected automatically to obtain complete images of the bone and the implanted biomaterial at a magnification of $\times 40$.

From the histological images, the proportions occupied by bone, bone graft and soft tissue were identified using a digitizing tablet (Cintiq Companion, Wacom, Germany), colored (Photoshop, Adobe, San Jose, CA, USA), and digitally measured using an image analysis program (CallSens, Olympus, Tokyo, Japan).

### 2.6. Statistical Analysis

The results of the microtomographic and histometric analysis are reported as mean and standard deviation values (n = 10). The differences between groups were statistically analyzed using a paired Student's *t*-test for each of the variables. *p*-values < 0.05 were considered to be statistically significant. Data were represented in box plots (minimum, 25% percentile, median, 75% percentile, maximum).

## 3. Results

### 3.1. Shark Tooth-Derived Bone Graft Physicochemical Characterization

ICP-OES was used to determine the elemental composition of the shark tooth-derived bone graft and identified a calcium phosphate compound with the presence of minor elements such as sodium (Na) (1.25 ± 0.02 wt%) and magnesium (Mg) (0.54 ± 0.02 wt%), as well as other trace elements such as strontium (Sr), potassium (K), aluminum (Al), and iron (Fe). Ion chromatography also detected the presence of a relevant amount of fluorine (F) (1.05 ± 0.11 wt%).

(Fourier-transform) FT-Raman analysis was carried out to evaluate the composition of the calcium phosphate in terms of functional groups (Figure 1A). Raman peaks were detected at 1070, 960 and 430 cm$^{-1}$, attributed respectively to asymmetric stretching, symmetric stretching, and symmetric bending of $PO_4^{-3}$ groups [19]. A peak at 588 cm$^{-1}$ attributed to asymmetric bending of $PO_4^{-3}$ was also detected. Moreover, asymmetric stretching of $CO_3^{-2}$ groups was detected at 1064 cm$^{-1}$ [19]. The absence of the Raman bands associated with organic compounds from the original raw material, including those attributed to amide III (1271 cm$^{-1}$), C-H$_2$ groups of lipids and some proteins (1450 cm$^{-1}$), and amide I (1665 cm$^{-1}$), proves the efficiency of the thermal processing to obtain anorganic compounds [20,21].

The structural evaluation of the calcium phosphate, performed by X-ray diffractometer, found the characteristic X-ray diffraction pattern of a crystalline material (Figure 1B). The calcium phosphate structure was established on a combination of apatitic and non-apatitic phases. In the case of apatitic phases, peaks attributed to an apatite-(CaOH) (including hydroxyapatite, $Ca_5(PO_4)_3OH$ and apatite-(CaF) ($H_{0.6}Ca_5F_{0.4}O_{12.6}P_3$)) and a fluorapatite

($Ca_5(PO_4)_3F$) were found in a semi-quantified percentage of around 70% of the total composition. The non-apatitic phases were attributed to tricalcium bis (orthophosphate) ($Ca_3O_8P_2$) and whitlockite ($Ca_{18}Mg_2H_2(PO_4)_{14}$) in a semi-quantified percentage of around 30% [17,18].

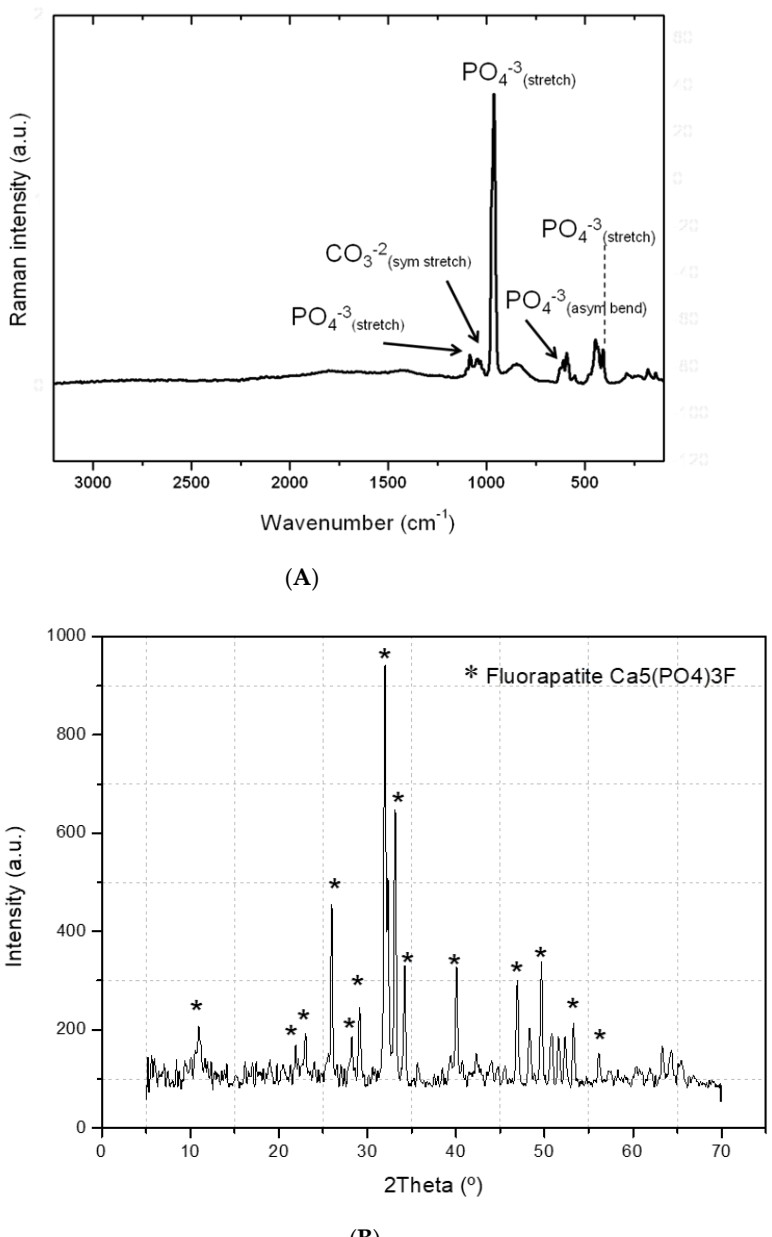

**Figure 1.** (Fourier-transform)FT-Raman spectra (**A**) and X-ray diffractogram pattern (**B**) of calcium phosphate derived from shark tooth (*P. glauca* and *I. oxyrinchus*).

### 3.2. Histological and Histometric Analyses

For the histological assessment, semi-quantitative evaluation techniques were used to quantify the different inflammatory cells, as well as the formation of new blood vessels and the presence of fibrosis and fatty infiltrate.

Optical images of shark tooth-derived bone graft and Bio-Oss® samples show that macroscopically both specimens presented a normal anatomy, with no signs of bone alteration (Figure 2). Normal bone tissue is seen in contact with both types of bone grafts. Inflammation and fibrosis were not considered significant or incompatible with the osseointegration of both grafts. No changes in tissue morphology were observed. Also, no

inflammatory cells, traumatic necrosis, foreign debris, fatty infiltrate, or granulomas were observed, which demonstrates high biocompatibility in both the tested and control bone grafts.

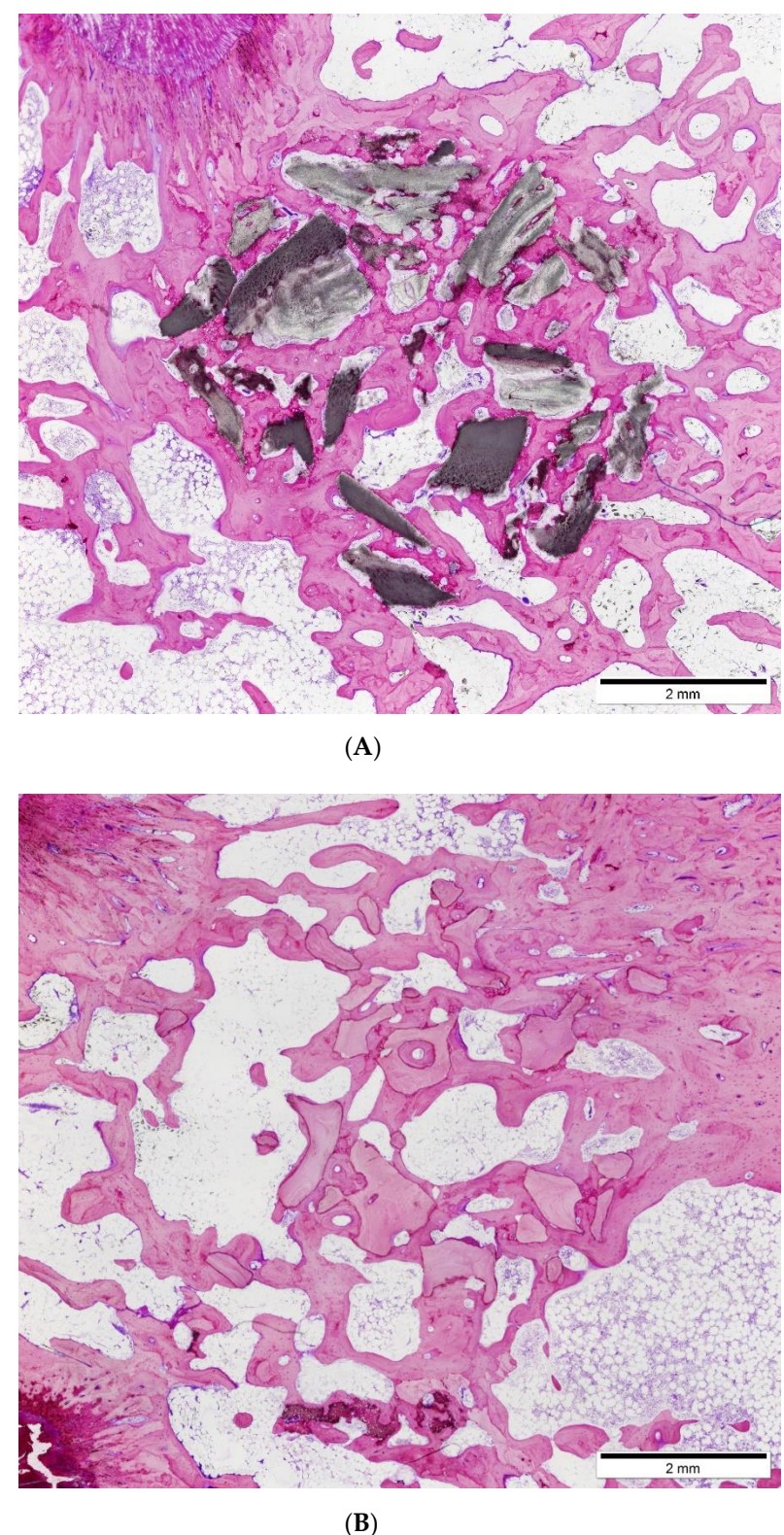

(**A**)

(**B**)

**Figure 2.** *Cont.*

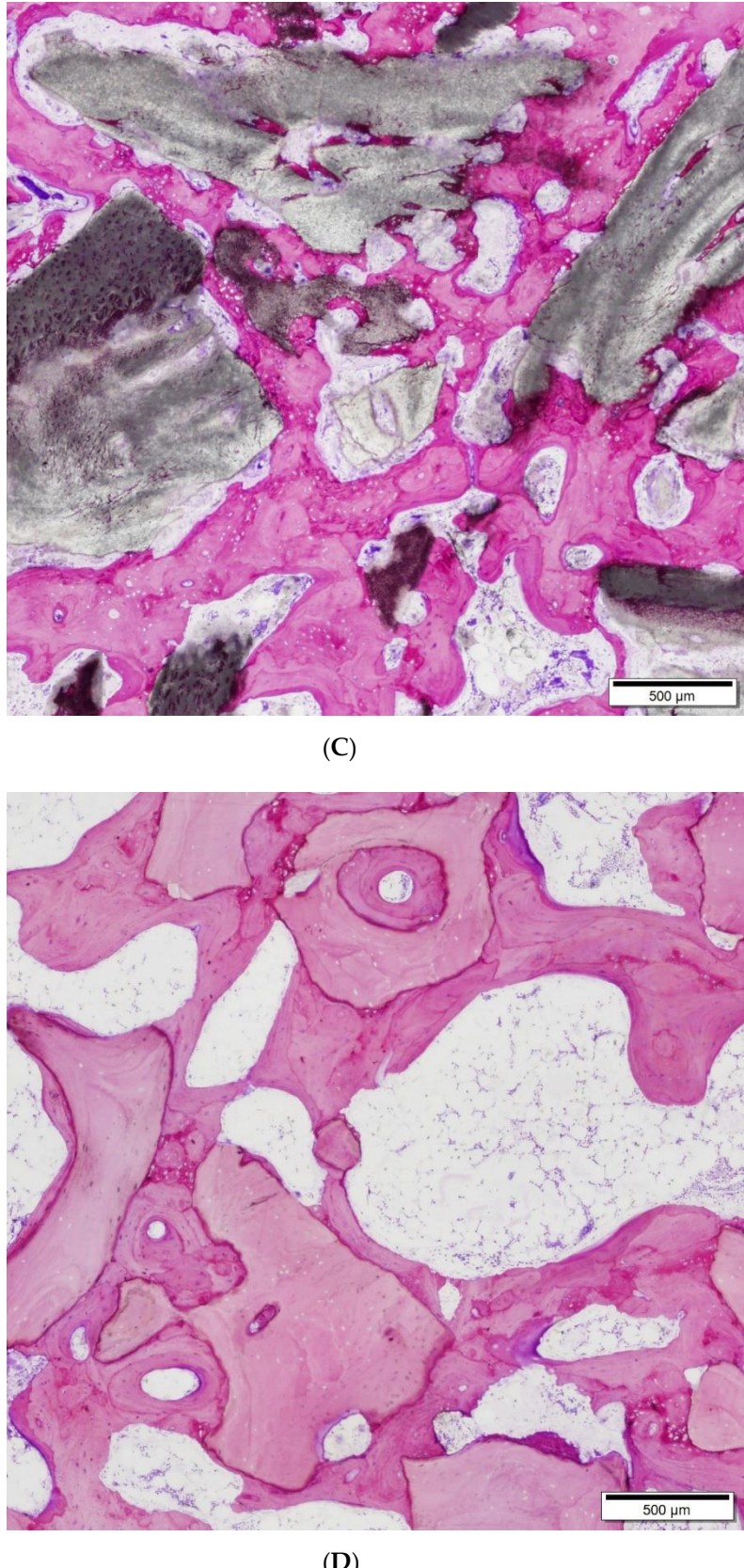

(**C**)

(**D**)

**Figure 2.** Optical images of shark tooth-derived bone graft (**A,C**) and Bio-Oss® (**B,D**). Original magnification 4× (**A,B**), and higher magnification (**C,D**).

The most relevant semi-quantitative histological data are depicted in Table 1. Cellular observations were quantified by counting the number of various cells per high-power field. The cellular response (presence of polymorphonuclear cells, giant cells, plasma cells, lymphocytes, and macrophages) in the test sample has a score of 0 or 1, denoting respectively the absence or presence of fewer than 5 per high-power field. No necrosis was observed in the test sample. Fibrosis (1 moderately thick band and 2 narrow bands) was observed in 3 of the 10 test samples (4R, 5R and 6L) but was not statistically significant. Information on vascularity was obtained by counting the number of capillaries per high-power field ($400\times$). There was no evidence of neovascularization or fatty infiltrate. With the data shown here we can conclude that, under the conditions of our study, the shark tooth-derived bone graft is as "non-irritant" to the tissue as the Bio-Oss® control sample.

**Table 1.** Semi-quantitative histological evaluation at 12 weeks post implantation in the test group (shark tooth-derived bone graft) and in the control group (Bio-Oss®).

| | Test Sample | | | | | | | | | | Control Sample | | | | | | | | | |
|---|---|---|---|---|---|---|---|---|---|---|---|---|---|---|---|---|---|---|---|---|
| Animal Number | 1R | 2R | 3R | 4R | 5R | 6L | 7L | 8L | 9L | 10L | 1L | 2L | 3L | 4L | 5L | 6R | 7R | 8R | 9R | 10R |
| Inflammation Polymorphonuclear | 0 | 0 | 0 | 0 | 0 | 0 | 0 | 0 | 0 | 0 | 0 | 1 | 0 | 0 | 0 | 0 | 0 | 0 | 0 | 0 |
| Lymphocytes | 0 | 0 | 0 | 0 | 0 | 0 | 0 | 0 | 0 | 0 | 0 | 0 | 0 | 0 | 0 | 0 | 0 | 0 | 0 | 0 |
| Plasma cells | 0 | 0 | 0 | 0 | 0 | 0 | 0 | 0 | 0 | 0 | 0 | 0 | 0 | 0 | 0 | 0 | 0 | 0 | 0 | 0 |
| Macrophages | 1 | 1 | 1 | 1 | 1 | 1 | 0 | 1 | 1 | 0 | 2 | 2 | 0 | 0 | 0 | 0 | 0 | 0 | 0 | 1 |
| Giant cells | 0 | 1 | 1 | 0 | 0 | 1 | 0 | 0 | 0 | 0 | 0 | 1 | 0 | 0 | 0 | 0 | 0 | 0 | 0 | 0 |
| Necrosis | 0 | 0 | 0 | 0 | 0 | 0 | 0 | 0 | 0 | 0 | 0 | 0 | 0 | 0 | 0 | 0 | 0 | 0 | 0 | 0 |
| Sub-Total ($\times 2$) | 1 | 2 | 2 | 1 | 1 | 2 | 0 | 1 | 1 | 0 | 2 | 4 | 0 | 0 | 0 | 0 | 0 | 0 | 0 | 1 |
| Neovascularization | 0 | 0 | 0 | 0 | 0 | 0 | 0 | 0 | 0 | 0 | 0 | 1 | 0 | 0 | 0 | 0 | 0 | 0 | 0 | 0 |
| Fibrosis | 0 | 0 | 0 | 2 | 1 | 1 | 0 | 0 | 0 | 0 | 3 | 2 | 0 | 0 | 0 | 0 | 0 | 0 | 0 | 2 |
| Fatty infiltrate | 0 | 0 | 0 | 0 | 0 | 0 | 0 | 0 | 0 | 0 | 0 | 0 | 0 | 0 | 0 | 0 | 0 | 0 | 0 | 0 |
| Sub-Total | 0 | 0 | 0 | 2 | 1 | 1 | 0 | 0 | 0 | 0 | 3 | 3 | 0 | 0 | 0 | 0 | 0 | 0 | 0 | 2 |
| Total | 1 | 2 | 2 | 3 | 2 | 3 | 0 | 1 | 1 | 0 | 5 | 7 | 0 | 0 | 0 | 0 | 0 | 0 | 0 | 3 |
| Group Total | 15/10 = 1.5 | | | | | | | | | | 15/10 = 1.5 | | | | | | | | | |
| Average | TEST (-) CONTROL = 1.50 − 1.50 = 0.00 (0) | | | | | | | | | | | | | | | | | | | |
| Traumatic necrosis | 0 | 0 | 0 | 0 | 0 | 0 | 0 | 0 | 0 | 0 | 0 | 0 | 0 | 0 | 0 | 0 | 0 | 0 | 0 | 0 |
| Foreign debris | 3 | 3 | 3 | 3 | 3 | 3 | 3 | 3 | 3 | 3 | 3 | 3 | 3 | 3 | 3 | 3 | 2 | 2 | 2 | 3 |
| No. sites examined | The whole contour of the biomaterial | | | | | | | | | | | | | | | | | | | |

In relation to the histometric evaluation, Figure 3 shows typical optical images of the shark tooth-derived bone graft samples before and after the digital coloring and analysis. After coloring, the image analysis program automatically detected the areas of color corresponding to the identification of bone tissue, biomaterial, and connective tissue. From this value, the percentages of bone graft and bone tissue were calculated (Figure 4). As observed, shark tooth-derived bone graft samples showed higher density of bone graft (biomaterial) and bone tissue in the volume of interest ($43 \pm 6\%$ and $38 \pm 5\%$) when compared with the control group ($28 \pm 5\%$ and $30 \pm 6\%$). The cause of these differences in bone graft density could be due to the grain size (0.5–1 mm in the test and 0.25–1 mm in the control), which generated a larger occupied area of shark tooth-derived bone graft. The difference in the bone percentages may indicate that shark tooth-derived bone graft allows osteoconduction to take place more effectively. Both differences were statistically significant.

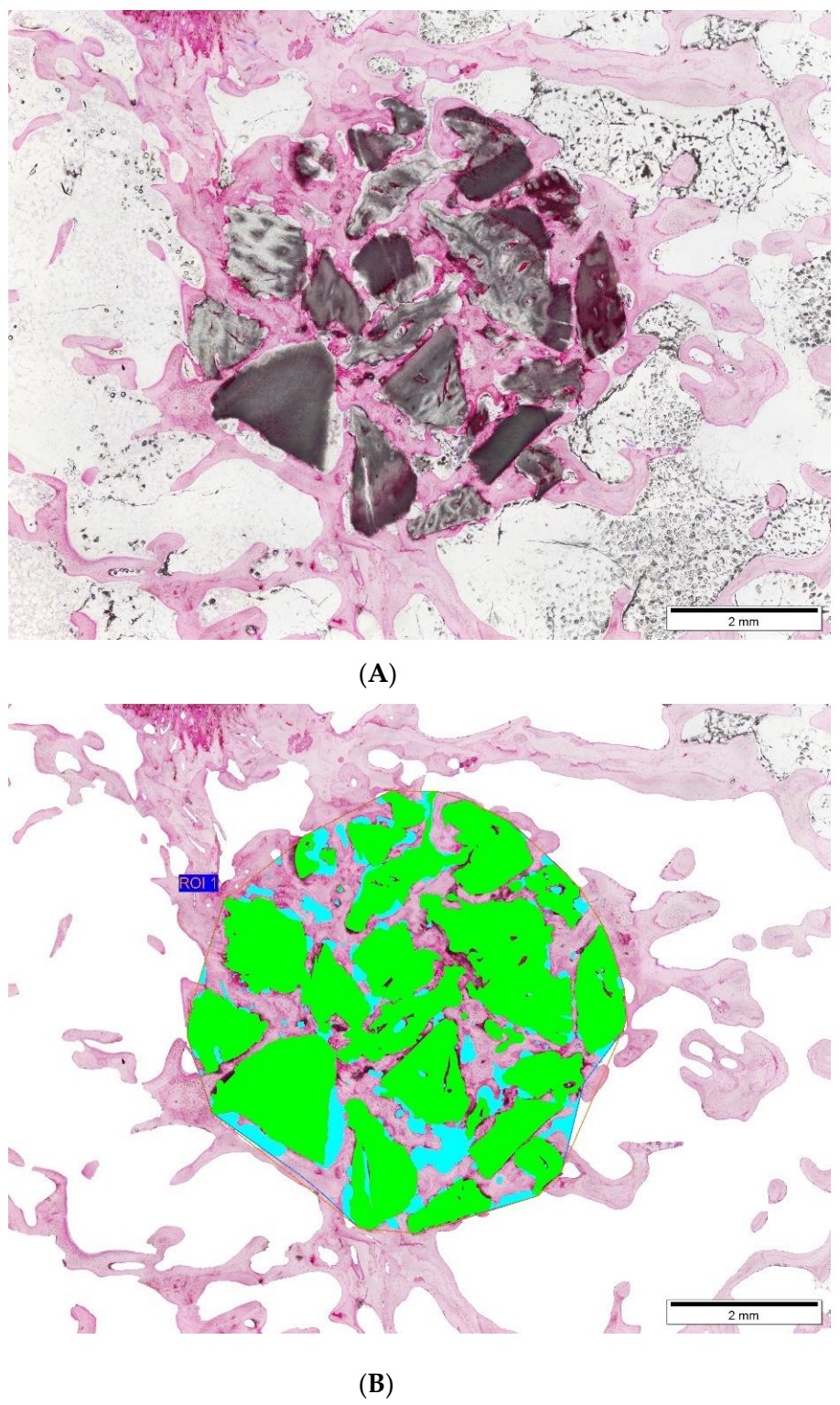

(**A**)

(**B**)

**Figure 3.** Shark tooth-derived bone graft samples before (**A**) and after (**B**) digital coloring and analysis: bone graft (green), bone tissue (pink) and connective tissue (blue). Magnification 4×.

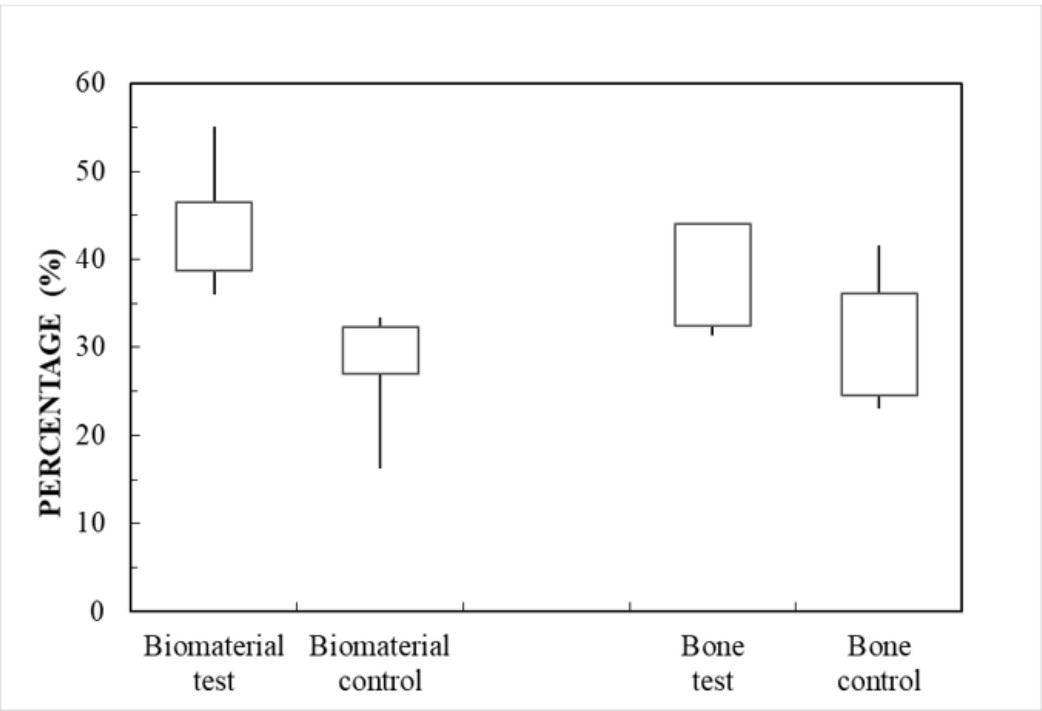

| | Test | Control | *p* |
|---|---|---|---|
| Bone graft–biomaterial (%) | 43 ± 6 | 28 ± 5 | <0.001 |
| Bone tissue (%) | 38 ± 5 | 30 ± 6 | 0.003 |

**Figure 4.** Boxplot and table of the histomorphometric results. Percentage of bone graft (biomaterial) and bone tissue for the test (shark tooth-derived bone graft) and the control (Bio-Oss®).

### 3.3. Microtomographic Evaluation

The micro-computed tomography analysis found the percentage area occupied by the bone tissue, the bone graft + bone tissue, and the intersection percentage of the bone graft with bone tissue (Figure 5). These values were carefully analyzed due to the difficulty in detecting the control material in the tomographic image, because the Bio-Oss® presented a density similar to that of bone tissue. In both cases, the percentage of bone graft occupancy was similar: 34 ± 7% for the test and 28 ± 7% for the control. Nevertheless, there was a significant increase in the area occupied by the bone tissue in the test samples (54 ± 6%) in relation to the control samples (27 ± 8%). Moreover, the percentage of intersection bone graft-bone tissue seems to be favored by the shark tooth-derived bone graft, reaching very high values (86 ± 8%) when compared with the control (30 ± 1%).

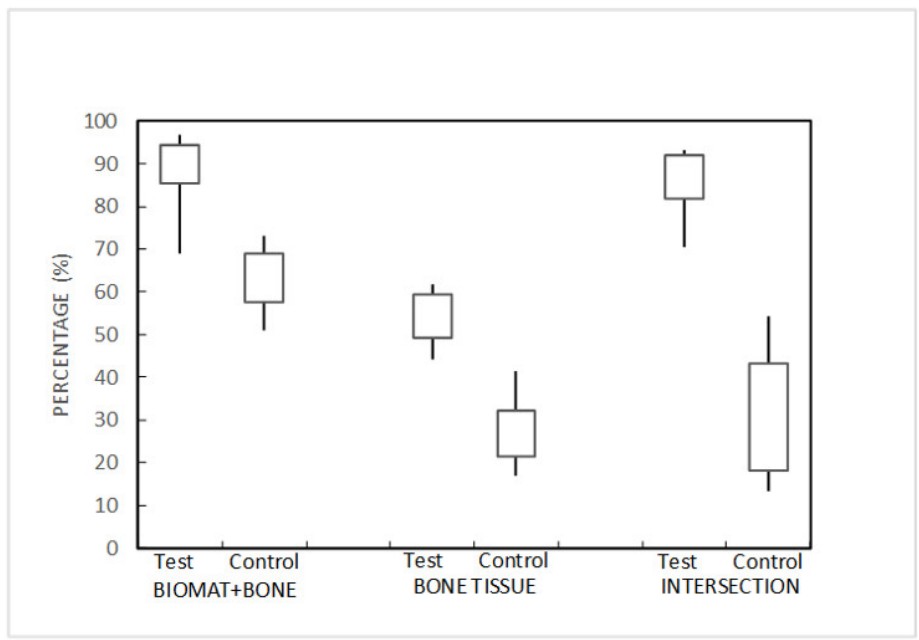

| | Test | Control | $p$ |
|---|---|---|---|
| Bone graft + Bone tissue (%) | 89 ± 8 | 62 ± 7 | <0.001 |
| Bone graft (%) | 34 ± 7 | 28 ± 7 | 0.031 |
| Bone tissue (%) | 54 ± 6 | 27 ± 8 | <0.001 |
| Intersection (%) | 86 ± 8 | 30 ± 1 | <0.001 |

**Figure 5.** Boxplot and table of the microtomographic results. Percentage of the bone graft + bone tissue (BIOMAT + BONE), the bone tissue, and the intersection in percentage for the marine bone graft (test) and the Bio-Oss® (control).

## 4. Discussion

Research into shark tooth-derived calcium phosphate compounds with traces of F, Na and Mg ions marks a recent development in the treatment of bone defects [17]. The crystalline structure of this bone graft is a combination of an apatitic phase (65–70%), including hydroxyapatite and fluorapatite, and a non-apatitic phase (35–30%), mainly whitlockite.

Preliminary in vitro studies have shown that this biphasic marine bone graft promotes the proliferation of MC3T3 preosteoblasts after 21 days of incubation and results in a significant increase in osteogenic activity ($p < 0.01$) [17]. Moreover, in vivo assays in a rodent model demonstrated an excellent osseointegration grade 3 weeks after implantation, with significantly higher bone mineral density ($p < 0.05$) than commercial synthetic HA/TCP bone graft (60%/40%) [18], and potential osteoconductive properties, due to the porous morphology and the formation of intergranular cavities which allowed the ingrowth of new bone tissue cells.

Biomaterials for clinical applications are subject to the strict requirements set out in ISO 10993-6:2007 (E), which specifies "test methods for the assessment of the local effects after implantation of biomaterials intended for use in medical devices". The purpose of such tests is to establish the safety and efficacy of the biomaterial in a biological environment, to ensure that it does not generate adverse reactions or release toxic substances.

Our study demonstrated the biocompatibility of a marine-origin bone graft obtained from shark teeth. The preclinical trial carried out on critical femoral defects in rabbits showed a non-irritant behavior and revealed that the test group (marine-origin bone graft) presented greater osseointegration at 12 weeks than the control group (bovine-origin bone graft). Results from the histological analysis showed a higher area of occupancy by bone

tissue in the test material (38 ± 5%) than in the control (30 ± 6%). Additional evaluation by micro-CT analysis confirmed this finding, given that values for bone mineral density were statistically higher in the test group (54 ± 6%) than in the control group (27 ± 8%). Moreover, the measurement of the bone graft-bone tissue intersection was much higher for the shark tooth-derived bone graft (86 ± 8%) than the control (30 ± 1%). While these figures confirm the superiority of the test bone graft in terms of bone formation and intersection measurements, the actual microtomographic values are likely to be overestimated, probably due to the similar radiological characteristics of the control sample and the bone tissue. This fact would also explain the large differences in the percentage of intersection. Nevertheless, taken as a whole, the results of the histometric and microtomographic analysis support the conclusion that the specific composition and structure of this marine-origin bone graft plays a key role in the promotion of bone regeneration.

The presence of fluor in the form of fluorine in the marine bone graft composition participates in the biochemical processes that enhance the synthesis of growth factors in bone cells, contributing to the differentiation of osteoprogenitor cells [22] and also to bone consolidation and regeneration [23]. In the form of fluorapatite, this element promotes a more crystalline and therefore more stable apatite [24], improving the mechanical properties with respect to hydroxyapatite. The non-apatite phase, on the other hand, is less crystalline and therefore less stable, allowing some degree of resorption.

The incorporation of Mg in the form of whitlockite is also of interest because of its influence on the synthesis of parathyroid hormone, which regulates bone homeostasis [25]. Furthermore, the presence, albeit to a lesser extent, of other ions such as Na, Sr and K contributes to cell adhesion and decreases osteoclastic activity [26].

Our findings further confirm shark tooth-derived bone graft as a good alternative to other bone xenografts on the market. Its clinical applications are not only limited to small bone defects, namely dental surgeries, but can be extended to the field of orthopedic and traumatological surgery, where there are clinical situations requiring high amounts of bone graft for the treatment of critical-sized bone defects. Future clinical assays would be necessary to assess its application in larger bone defects.

## 5. Conclusions

This preclinical evaluation demonstrates the biocompatibility and suitability of shark tooth-derived bone graft for the treatment of critical-sized bone defects. Under the conditions of this study, the test bone graft exhibited an adequate biological response, with no signs of inflammatory foreign body reactions, fibrosis, or necrosis in any of the cases 12 weeks after implantation.

Results from histological techniques and microtomographic analysis revealed good osseointegration, with bone mineral density and percentage of intersection values superior to the bovine-derived bone graft used as a control.

The biphasic structure, with both an apatitic (65–70%) and non-apatitic (35–30%) phase, as well as the specific composition of the shark tooth-derived bone graft, mainly calcium phosphates enriched with F and Mg, may be important factors stimulating bone regeneration.

**Author Contributions:** R.O.-P., P.G., J.S. and F.M.M. contributed to the acquisition, analysis, and interpretation of data, drafting of the paper, approval of the submitted, and final version. R.O.-P., M.P., E.L.-S., M.L.-Á., M.L.-P., J.S. and P.G. contributed to the research design, interpretation of data, and revising the paper and approval of the submitted and final versions. R.O.-P., M.P., A.G.-C. and F.M.M. carried out the experiment and contributed to the acquisition and analysis of data. All authors have read and agreed to the published version of the manuscript.

**Funding:** This research was funded by Xunta de Galicia, grants IN855A 2016/06 (Programa Ignicia–GAIN), ED431C 2017/51 (Competitive Reference Groups) and ED431D 2017/13 (Research networks); and by European Union Interreg Programs, projects IBEROS (0245_IBEROS_1_E, POCTEP 2015), CVMar+i (0302_CVMAR_I_1_P, POCTEP 2015) and BLUEHUMAN (EAPA_151/2016, Atlantic Area 2016).

**Institutional Review Board Statement:** The study was conducted according to the guidelines of the Declaration of Helsinki, and approved by Ethics Committee of Galicia (protocol code 03/18/LU-002 and date of approval 2018/02/09).

**Informed Consent Statement:** Not applicable.

**Data Availability Statement:** Not applicable.

**Acknowledgments:** This research was financially supported by Xunta de Galicia funds, Grants IN855A 2016/06 (Programa Ignicia–GAIN), ED431C 2017/51 (Competitive Reference Groups) and ED431D 2017/13 (Research networks). Financial support from European Union Interreg Programs by projects IBEROS (0245_IBEROS_1_E, POCTEP 2015); CVMar+i (0302_CVMAR_I_1_P, POCTEP 2015) and BLUEHUMAN (EAPA_151/2016, Atlantic Area 2016) is gratefully acknowledged.

**Conflicts of Interest:** The authors declare no conflict of interest.

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
