# Peer review of "Preclinical Evaluation of an Innovative Bone Graft of Marine Origin for the Treatment of Critical-Sized Bone Defects in an Animal Model"

_applsci, doi:10.3390/app11052116_

Round 1
Reviewer 1 Report
Few suggestion/comments:
- It is good to clearly make a difference between cancellous bone and cortical bone in the manuscript.
- when talking about the improved mechanical properties (e.g. page 14, paragraph 2, line 7), it should be clearly mentioned which mechanical properties are improved? Flexural strength/ Fracture toughness etc. ? How much?
- In the section 5 (conclusion) or maybe in the section 1 last paragraphs (introduction), the essential necessity of mechanical tests for presented bone graft in the manuscript should be mentioned/suggested. Biocompatibility is not the only factor!
Reviewer 2 Report
General comments. The use of sharks teeth as a source of calcium phosphate for bone grafts is a potentially attractive proposition so long as their consistent composition can be ensured and standardised. This article provides a convincing demonstration of their compatibility in this pre-clinical study. The article is clear and well written and I only have a few minor comments and suggestions for improvements.
Abstract
The abstract is a well-written summary of the article.
Introduction
The introduction gives a useful background to the work. There are a few minor errors:-
Paragraph 2.When referring to Masquelet’s method you should say “more recently” rather than “recently” because the reference is 2000, which is hardly very recent (though of course it is in comparison with 1950).
Page 2. There is no need for “therefore” in the first sentence of the second paragraph as this is a separate point and does not follow on.
Paragraph 3. “… non-human species as received much…” “as” should be “has”
Materials and Methods
The methods are generally well-described.
Why are teeth from 2 different species mixed together to make the granules? Wont this potentially alter the composition, depending on the relative proportions of each one in different batches?
There seems to be some contradiction in the placement of the biomaterial in the distal femur in Page 4 paragraph 3 and paragraph 6. In paragraph 3 you say that the materials were assigned randomly to both the left and right sides, but in P 6 you seem to say the placements were not random, which is correct, or please clarify?
P 6. Can you specify which type of cells were scored to give the irritant index? Is this method a standard one? If so, please provide a reference.
Results
These are clearly presented.
Fig 3: Is there a reason why you have only shown a representative illustration of the digital colouring and analysis for a shark tooth sample and not included a BioOss sample? It would be nice to have one of these (before and after) as well for comparison.
Discussion
Is there any evidence from the in vitro studies that shark tooth material is able to be resorbed by osteoclasts? If so, it would be good to include references to it on page 14 where you are discussing the presence of whitlockite in potentially promoting resorption but then say that Na, Sr and K may inhibit it. Do you envisage that the material might be used as a permanent or temporary bone substitute?
Conclusions. Last sentence. I don’t think you have sufficient evidence in this study to say that the specific composition and presence of ions were important factors – you could say that they may be.
Reviewer 3 Report
The study uses teeth from one endangered shark species. Not sure if it can be used as bone graft due to the origin of the biomaterial. The in vivo analysis looks poorly done in micro ct and histology, doesn't have statistical analysis and the interpretation in results and discussion is minimum. All comments are list as below :- English need to be edited by professional editor
- On the International Union for the Conservation of Nature (IUCN) Red List. Included under “Endangered” short-fin mako (Isurus oxyrinchus). Authors wrote the study like if they can recycle the unused teeth from consumers. How do they know that the use of these teeth as raw biomaterial for bone regeneration will not increase the fishing of these animal? Pacoureau, N., Rigby, C.L., Kyne, P.M. et al. Half a century of global decline in oceanic sharks and rays. Nature 589, 567–571 (2021).
- Abstract: missing results (percentage of new bone formation and remaining graft with statistical analysis.)
- …Xenografts of bovine origin has given good results in dental and maxilofacial surgeries, but contradictory results have been achieved in orthopedic surgery, in particular foot and ankle surgery [10,11]… Please develop the idea of the contradictory results from references 10 and 11. Modify reference 11 because is not about Xenografts of bovine origin.
- Please develop the idea of why are you using Bio Oss, because is not clear
- Last paragraph of the introduction shows partially results and methodology, please locate it in material and methods and results, accordingly.
- Material and methods doesn’t show the Prionace glauca and Isurus oxyrinchus species teeth ratios in which were mixed to create the new bone graft. How were removed the teeth from the mandible and maxilla? How do they know both types can induce bone regeneration?
- 4. Micro-computed tomography indicates volume of interest (VOI) but only indicates 6mm of diameter but not the height. Did the authors only did 2D analysis? And should be Region of interest, please clarify. Also indicate which unit was used to analyze the percentage of biomaterial.
- Was the same VOI 2 use in all samples?. Because if the are different sizes this could lead to a manipulation in the results of new bone formation. Please clarify
- Slightly describe each one of test done for physicochemical characterization (2.1. Shark tooth-derived bone graft processing method).
- Figure 2: indicate the right scale bar. Looks like the defects are around 3mm. Not possible to determine ‘osseointegration neither o inflammatory cells’ with that magnification, please add a higher zoom image.
- Figure 3: indicate the right scale bar. Looks like the defects are around 3mm.3B indicate ROI, in material and methods was said that was VOI, please clarify.
- Results…As observed, shark tooth-derived bone graft samples showed higher density of bone graft (biomaterial) and bone tissue in the volume of interest (43 ± 6% and 38 ± 5%) when compared with the control group (28 ± 5% and 30 ± 6%)…. Is there statistically significant difference?
- 3. Microtomographic evaluation: In both cases, the percentage of bone graft occupancy was similar: 34 ± 7% for the test and 28 ± 7% for the control. Nevertheless, there was a significant increase in the area occupied by the bone tissue in the test samples (54 ± 6%) in relation to the control samples (27 ± 8%). Is there statistically significant difference?
- Discussion: lacks comparison between own new bone formation results with other studies.
Round 2
Reviewer 3 Report
The authors revised the manuscript by reviewer’s opinions. I suggest to accept this manuscript for publication .
